# Immunomodulation in Cystic Fibrosis: Why and How?

**DOI:** 10.3390/ijms21093331

**Published:** 2020-05-08

**Authors:** Vincent D. Giacalone, Brian S. Dobosh, Amit Gaggar, Rabindra Tirouvanziam, Camilla Margaroli

**Affiliations:** 1Department of Pediatrics, Emory University School of Medicine, Atlanta, GA 30322, USA; vgiacal@emory.edu (V.D.G.); brian.seth.dobosh@emory.edu (B.S.D.); 2Center for CF & Airways Disease Research, Children’s Healthcare of Atlanta, Atlanta, GA 30322, USA; 3Department of Medicine, University of Alabama at Birmingham, Birmingham, AL 35233, USA; agaggar@uabmc.edu (A.G.); cmargaroli@uabmc.edu (C.M.); 4Pulmonary Section, Birmingham VA Medical Center, Birmingham, AL 35233, USA

**Keywords:** cystic fibrosis, inflammation, immunotherapy, lung disease, proteases

## Abstract

Cystic fibrosis (CF) lung disease is characterized by unconventional mechanisms of inflammation, implicating a chronic immune response dominated by innate immune cells. Historically, therapeutic development has focused on the mutated cystic fibrosis transmembrane conductance regulator (CFTR), leading to the discovery of small molecules aiming at modulating and potentiating the presence and activity of CFTR at the plasma membrane. However, treatment burden sustained by CF patients, side effects of current medications, and recent advances in other therapeutic areas have highlighted the need to develop novel disease targeting of the inflammatory component driving CF lung damage. Furthermore, current issues with standard treatment emphasize the need for directed lung therapies that could minimize systemic side effects. Here, we summarize current treatment used to target immune cells in the lungs, and highlight potential benefits and caveats of novel therapeutic strategies.

## 1. Introduction

The study of cystic fibrosis (CF) lung disease has generated a wealth of knowledge on epithelial cell dysfunction and potential options for targeted therapeutics [1,2,3]. While these findings have provided key information needed for the development of modulator therapies directed at the cystic fibrosis transmembrane conductance regulator (CFTR), growing interest in the role of immune cells in CF, especially in the lungs, is advancing our understanding of previously unknown disease mechanisms. CFTR mutations have been associated with impaired pathogen clearance by myeloid cells [4,5,6,7,8], altered B-cell activation [9], and cytokine secretion by T-cells [10]. However, the exact role that CFTR plays in modulating key immune cell functions remains unclear, as recent studies have challenged early findings that CFTR deficiency impairs pathogen clearance by myeloid cells [11,12]. As such, the adaptability of immune cells, particularly neutrophils [13], to diseased microenvironments may be of greater importance than loss of CFTR regarding their role in CF. In addition, there is a great variety of CFTR mutations, with over 2000 identified mutations spread across seven classes. Each of these classes represents different mechanisms causing CFTR deficiency or dysfunction, and thus many opportunities for precision medicine [14]. However, no therapy to date has been able to address the onset of chronic inflammation in the CF lung [15]. New therapies directly addressing cellular mechanisms of inflammation, especially regarding neutrophils, are urgently needed. Neutrophils are the most abundant leukocyte subset in the human body in terms of new cells produced per day (approximately 10^9^ per kg) and an essential component of innate immunity for their role as highly efficient phagocytes and regulators of immune responses [16]. In CF patients, neutrophils are massively recruited to the airways and are major drivers of lung inflammation [17,18]. Among resident immune cells, tissue-resident macrophages are important for maintaining lung homeostasis [19], and another focal point of immune imbalance in CF airways [20].

Immunological dysfunction or reprogramming of innate cells is gaining significant attention as a contributing factor to the inability to control infections, either due to intrinsic or acquired defects. These infections include common CF pathogens such as *Staphylococcus aureus* and *Pseudomonas aeruginosa* [21], but also emerging and possibly more dangerous pathogens such as *Mycobacterium abscessus* [22]. Although innate immune cells have captured much of the growing interest in the immunology of CF, important discoveries have been made in adaptive immune cells, as well. T cells are heavily suppressed by neutrophil activity in CF lungs [23]. While this suppression may avoid autoimmune responses to self-antigens present in this chronically inflamed environment, it also be problematic by excluding regulatory subsets of T cells. Fewer studies have been conducted on the role of B cells in CF, but there is preliminary evidence for CFTR deficiency contributing to heightened B cell activation and development of lymphoid follicles [9]. These observations form a foundation from which to investigate in more depth the interplay between immune subsets in the CF lung, and identify key mechanisms of immunomodulation for therapeutic targeting.

## 2. Targeting Immune Cells in the CF Lung

### 2.1. Neutrophils

The progression of events in early CF lung disease is still under debate. Traditionally, bacterial infections established after mucus obstruction were viewed as the driving force for reactive inflammation, but recent reports proposed that inflammation may be initiated by early mucus plugging in the absence of infection [24], and reviewed in [25]. Although the exact sequence of events has not yet been defined, murine models have been used to demonstrate that hypoxia-induced necrosis of airway epithelia can initiate neutrophil influx via the interleukin (IL)-1 receptor signaling in the absence of infection [24]. Neutrophilic inflammation quickly becomes a major factor in the pathogenesis of CF lung disease due to acquisition of a pathological phenotype. CF airway neutrophils exhibit exocytosis of primary and secondary granules, loss of phagocytic receptors, and metabolic reprogramming that contributes to delayed apoptosis [26,27]. These phenotypic shifts create a critical role for neutrophils in the CF lung environment, which has been reviewed extensively [17,18,28,29]. Although mechanisms explaining how neutrophils acquire this phenotype are still being investigated, it is readily apparent how much it contributes to disease progression.

Exocytosis of the primary granules releases the serine protease neutrophil elastase (NE) into the extracellular environment. NE causes degradation of connective tissue in the lung [30], promotes mucus production in the airways [31], and is capable of degrading CFTR [32]. Furthermore, activity of soluble NE in the bronchoalveolar lavage fluid from infants with CF can serve as a reliable predictor of bronchiectasis [33]. NE can be captured on the cell surface following release or become associated with exosomes, enabling it to resist inhibition by tissue anti-proteases and maintain catalytic activity [34,35]. These attributes demonstrate the therapeutic potential of targeting NE as a way to reduce inflammation and lung damage, which has indeed been a topic of discussion [36,37]. Importantly, inhibition of NE has been shown to promote wound healing in primary airway cells from CF patients [38]. It is important to note that NE is not the only protease implicated in the progression of CF lung disease. The matrix metalloproteinase (MMP) family has also been studied for its role in tissue degradation and as potential therapeutic targets [39].

In addition to proteases, other neutrophil granule proteins play important roles in CF lung disease. Myeloperoxidase (MPO), also contained in neutrophil primary granules, generates potent oxidants including hypochlorous acid [40]. The presence of MPO in the bronchoalveolar lavage fluid of patients with CF has been found to strongly correlate with development of bronchiectasis [41], as does methionine sulfoxide, a by-product of MPO activity [42]. Neutrophil degranulation has also been identified as a possible source of elevated resistin in CF patient sputum. As a potent inducer of neutrophil recruitment via ligation of toll-like receptor (TLR)-4, resistin may serve as a mechanism by which neutrophils exponentially increase their own recruitment to CF lungs as they degranulate [43].

The clear relationship between release of inflammatory mediators and progression of CF lung disease creates substantial need to modulate neutrophil exocytosis. Several compounds have been identified that can inhibit neutrophil granule exocytosis without impeding normal antimicrobial function by specifically targeting granule docking proteins [44,45] (Figure 1a). However, because CF patients are at high risk for contracting respiratory infections, it is essential that any suppression of neutrophil activity not leave patients vulnerable to common pathogens.

### 2.2. Eosinophils

Eosinophils are a class of granulocytes primarily involved in defense against parasites, but they can fulfill diverse roles including responding to fungal and viral infections as well as interfacing with the adaptive immune system [46]. Recent studies have reaffirmed that eosinophils, which have a more prominent inflammatory role in asthma [47], typically are not a driving force in CF lung pathology [48]. However, research into eosinophil activation [49] and their contribution to comorbidities such as allergic bronchopulmonary aspergillosis (ABPA) [50], eosinophilic esophagitis [51], and nasal polyps [52] in CF patients suggest they do have clinical relevance.

As demonstrated by Koller et al., eosinophils may be more activated in CF airways even if they are not proliferating. In one of the earliest reports of activated eosinophils having a potential role in CF lung pathology, increased serum levels of eosinophil cationic protein (ECP), an eosinophil activation marker, were detected in CF patients compared to healthy control subjects [49]. Interestingly, this increase in sputum ECP levels was not linked to a change in peripheral blood eosinophil counts but correlated strongly with sputum ECP levels. This finding was confirmed by stimulating granulocytes ex vivo and demonstrating higher release of ECP from eosinophils of CF patients compared to those of healthy controls and bronchial asthma patients [53]. Since eosinophil activity was also shown to impact clinical variables in that study, it suggests that even though this rare cell population may not have increased prevalence in CF patients, their heightened activity may have clinical relevance. Such finer details are important to consider when planning anti-inflammatory treatments for patients, especially considering other factors such as sex. Indeed, significantly higher populations of both eosinophils and mast cells have been detected in female CF patients compared to their male counterparts and may therefore require different dosages of anti-inflammatory drugs [54].

The kinase inhibitor (R)-Roscovitine has recently been suggested as having potential efficacy in treating CF by reducing release of peroxidase from activated eosinophils [55] (Figure 1b). Eosinophil peroxidase has structural similarities to MPO [56], which is implicated in CF lung damage [18], but has both pro- and anti-inflammatory properties [56]. (R)-Roscovitine has the additional benefit of inducing eosinophil apoptosis. Without proper efferocytosis, this may however lead to secondary necrosis [57] and result in acute inflammation, so care must be taken to avoid driving the inflammatory response in an attempt to reduce it.

Another therapeutic that may have efficacy in counteracting eosinophil activation in CF patients is benralizumab. This monoclonal antibody blocks the IL-5 receptor and induces eosinophil apoptosis, showing efficacy in treating eosinophilic asthma [58]. A study of benralizumab in asthmatics by Pham et al. showed reduced serum eosinophil-derived neurotoxin and ECP following eosinophil depletion [59]. While CF patients were excluded from this study, it may have efficacy in CF patients who display markers of eosinophil activation [49,53].

Additionally, it has been reported that the eotaxin receptor CCR3 on eosinophils from CF patients with ABPA (CF-ABPA) is upregulated compared to patients without this comorbidity [60]. This finding complements an earlier report that methylprednisolone, which has been shown to reduce eosinophil chemotaxis [61] and degranulation [62], improves the condition of CF-ABPA patients [63]. Another therapy evaluated in CF-ABPA patients is the anti-immunoglobulin E (IgE) monoclonal antibody omalizumab. Early case reports using this antibody therapeutically demonstrated reduced exacerbations and improved lung function in CF-ABPA patients with eosinophilia, elevated total serum IgE, and elevated *Aspergillus*-specific IgE [64,65,66,67], but larger retrospective studies did not find such efficacy [68,69]. Due to the difficulty in treating established *Aspergillus* infections in CF patients, identifying infection early is crucial. In a study by Keown et al., peripheral eosinophilia and exhaled nitric oxide were significantly higher for CF patients with confirmed ABPA compared to those who were only sensitized, while serum ECP levels showed a positive trend [70].

### 2.3. Basophils

In addition to neutrophils and eosinophils, other innate immune cells have been studied in CF. Basophils are a rare granulocyte subset in the blood stream that have a potent role in allergic inflammation [71]. Their potential role in CF was observed as early as 1980, when CF patients with *P. aeruginosa* infections were shown to have greater histamine release by basophils compared to those without such infections [72]. The potency of this vasodilating agent in driving the inflammatory response, especially for CF patients, is demonstrated by its use in assessing bronchial hyperreactivity [73]. Similar to the observation that eosinophils in CF patients may be more activated but not more abundant [49,53], basophils in CF patients have been shown to be more active in releasing histamine but not more abundant compared to control subjects [74,75].

Also similarly to eosinophils, basophils have a role in CF-ABPA. Gernez et al. analyzed basophils from CF-ABPA patients and identified increased expression of the activation marker CD203c compared to non-sensitized CF patients and those without *Aspergillus* infection [60]. Katelari et al. were later able to identify an increase in expression of both CD203c and CD63 using the basophil activation test in CF-ABPA patients vs non-ABPA CF patients, highlighting the potential diagnostic value of this method [76]. Given the severe pathology seen in CF patients with ABPA, and the heightened activation of histamine-producing basophils, antihistamine treatments may warrant increased attention. Antihistamines including loratadine, cetirizine, and fexofenadine are common medications for CF patients with allergies and are more selective for histamine receptors than muscarinic receptors [77]. Still it is desirable to minimize the tendency of antihistamines to cause drying or thickening of mucus in the airways, as has been demonstrated for cyproheptadine hydrochloride [78]. Continued investigation into the role of basophils in CF-ABPA patients could improve treatments by accurately distinguishing allergic exacerbations in CF patients from those caused by bacterial infections, which drive a neutrophilic response [79].

### 2.4. Mast Cells

Mast cells, which are active in allergy and immunomodulation, harbor numerous homeostatic roles such as wound healing and angiogenesis, are involved in early initiation of inflammation through secretion of vasodilators like histamine [80]. Given their wide-ranging abilities to modulate immune responses, a greater understanding of mast cell function in complex inflammatory diseases like CF could yield new strategies for improving treatment options. In a study of mast cells in lung disease, Andersson et al. observed a reduction in total mast cell density in the small airways of CF patients. This was attributed to decreased density of the mucosal mast cell subpopulation associated with healthy lung function while the connective tissue mast cells were unchanged. However, production of IL-6 by mast cells was increased in CF patients [81]. IL-6 has been shown to block the production of regulatory T cells (Tregs) normally induced by tumor growth factor-β (TGF-β) [82], and due to its role in maintaining the balance between Tregs and T helper 17 (Th17) cells [83], could be important for Th17-related pathology in CF. Since the proportion of connective tissue mast cells in small airways correlated negatively with lung function, mast cell-derived IL-6 could be a target for inhibition in CF patients, especially considering successful use of IL-6 trans-signaling blockade in a mouse model of pulmonary fibrosis [84]. Another cytokine target relevant to mast cell activity is IL-9. In a recent study by Moretti et al., IL-9 was shown to promote production of IL-2 by mast cells, leading to expansion of type 2 innate lymphoid cells and subsequent activation of inflammatory Th9 cells. Imatinib, a tyrosine kinase inhibitor that reduces IL-9-mediated lung mastocytosis, reduced mastocytosis and production of inflammatory cytokines in a CF mouse model [85] (Figure 1c).

Modulation of mast cell activity may be of special benefit to CF patients with nasal polyps. Polyps from CF patients have been shown to have elevated mast cell numbers and greater degranulation compared to those from non-CF individuals [86]. It may also be beneficial to promote autophagy in mast cells, as inhibition of this process attenuated bacterial killing by mast cells while induction by rapamycin promotes killing of *P. aeruginosa* [87]. Mast cell modulation may also be beneficial in *Aspergillus* infections. Production of the mast cell protease tryptase was decreased in patients with *Aspergillus*-specific IgE compared to IgE- patients [88]. As recently reviewed by Piliponsky et al., mast cells can effectively respond to *Candida* infections at mucosal surfaces [89]. Modulating mast cell activity in CF patients with *Aspergillus* infections may aid in controlling this opportunistic pathogen.

### 2.5. Macrophages

In addition to neutrophils, macrophages comprise a large proportion of CF airway leukocytes. Their major functions include clearance of debris and pathogens, promoting tissue remodeling, and regulating immune response [19]. Because of these crucial roles, the role of macrophages in CF lung disease has been studied extensively [19,20,90]. An important aspect of lung macrophages in CF is that like neutrophils [91], they exhibit defects in pathogen clearance despite acquiring highly pro-inflammatory phenotypes [20] (Figure 1d). The effect of CFTR deficiency on neutrophils has been studied [5,6], but has been described more in depth for macrophages in recent years [4,7,8]. These reports demonstrate clear evidence for deficiencies in CF macrophage function and suggest the opportunity for therapeutic intervention, including possible direct benefit to macrophage function from CFTR modulatory therapy. Although one of the primary roles of airway macrophages is to maintain a homeostatic environment, they have been shown to greatly increase production of pro-inflammatory cytokines in CF patients compared to those of healthy subjects [92]. A recent study by Lara-Reyna et al. suggests this may be due to metabolic reprogramming broadly affecting innate immune cells in CF. This study demonstrates that activation of the inositol-requiring enzyme 1 α pathway due to endoplasmic reticulum stress in macrophages increases glycolysis and drives production of pro-inflammatory cytokines including tumor necrosis factor α (TNFα) and IL-6 [93].

### 2.6. T Cells

While the airway lumen of CF patients experiences substantial influx of neutrophils, lymphocytes are actively recruited into the bronchial mucosa, and accumulate under the basement membrane [94]. Similar to neutrophils, T cells in CF patients demonstrate distinct pathological behavior. An important modulator of T cell function is the protease arginase 1 (Arg1), which was first evaluated in CF patients in 2005 by Grasemann et al. They identified an abundance of arginase in CF sputum samples which negatively correlated with lung function [95]. This enzyme is involved in the impairment of T cell signaling via downregulation of the CD3ζ chain [96]. Ingersoll et al. demonstrated the importance of Arg1 in negative regulation of T cell function in CF patients by identifying the link between Arg1 from CF airway fluid and suppression of T cell proliferation. Furthermore, they attributed much of this regulatory activity to neutrophils by demonstrating a positive correlation between expression of Arg1 on airway neutrophils and Arg1 activity in airway fluid [23]. The suppression of T cell activity may be contributing to the observation that CF patients have a significant increase in reported respiratory illness, even if they do not experience higher frequency of actual infection [97]. These findings suggest that inhibition of Arg1 activity, which has already shown promise in animal models of inflammatory disease [98,99], could have efficacy in treatment of CF. The importance of T cells in respiratory health is emphasized by a case report detailing a CF patient with mucosal-associated invariant T cell deficiency. Lung function was too high for considering a lung transplant, but the patient had exceptional risk for bacterial infections [100]. While this deficiency was likely attributed to an inherited defect independent from CF, it highlights the susceptibility of CF patients to immunological impairments.

Tregs are a crucial subset of T cells that regulate the immune response by maintaining self-tolerance and limiting inflammation [101]. Unsurprisingly, Treg dysfunction is implicated in CF. A reduction of CD4^+^CD25^high^FoxP3^+^ Tregs in CF patients was first identified in 2014, and the prevalence of this cell population was found to positively correlate with lung function as measured by forced expiratory volume in 1 s (FEV_1_) [102]. Chronic *P. aeruginosa* infections have been linked to reduced Treg counts in CF patients, especially those over the age of 16 [103]. As life expectancy continues to improve for CF patients, this problem may require therapeutic attention. One potential treatment relative to Tregs centers on indoleamine 2,3-dioxygenase (IDO) (Figure 2). This enzyme is involved in tryptophan degradation and when dysfunctional it has been attributed to an imbalance between Tregs and Th17 cells [104,105,106]. After demonstrating that IDO deficiency is associated with imbalanced Th17 and Treg populations in a murine model of CF, Ianitti et al. successfully corrected helper T cell populations and resolved inflammation by IDO restoration therapy. Although T cells are largely excluded from the airway lumen, Th17 T cells have been identified in the airway submucosa of CF patients in both early and established airway disease [107], and the Th17 pathway has been proposed to exert a major role in CF [105]. These findings suggest that correction of T cell imbalance in the CF lungs, especially Tregs for their anti-inflammatory role, could improve the course of airway inflammation.

### 2.7. B Cells

B lymphocytes form the other arm of the adaptive immune system and are also altered in CF. An earlier study by Sorensen et al. identified a higher frequency of plaque-forming cells, signifying differentiation into antibody-secreting cells, in CF patients who did not yet have severe pulmonary disease [108]. They noted normal plasma cell presence in patients with advanced disease and identified impaired B cell differentiation in cells activated in vitro. Similarly, Hubeau et al. found larger B cell-containing lymphoid aggregates in airways of CF compared to non-CF subjects, and similar presence of plasma cells in CF and non-CF tissue samples. They speculated that B cells rather than plasma cells are contributing to inflammation in CF airways, possibly due to their role as antigen-presenting cells [109]. It may be worth investigating if the influx of B cells with apparent lack of differentiation to plasma cells is a CF-related defect or possible control mechanism to prevent antibody-mediated inflammation. The latter could be explained by the resistance of CF B cells to treatment with dexamethasone. Indeed, while IgG production was found to be similar in CF and control patients, dexamethasone was demonstrated to enhance antibody production in stimulated control cells but had no effect on CF cells [110]. While CF and normal B cells were found to present staphylococcal superantigen to T cells in similar capacity, presentation by CF B cells was not inhibited by dexamethasone treatment [111]. This finding may indicate that a potential inflammatory role of CF B cells proposed by Hubeau et al. may be due to a resistance to acute inhibition of antigen presentation rather than an intrinsically heightened activity.

Consistently, an increased frequency of lymphocyte-containing tertiary lymphoid structures has been identified in the lungs of CF patients, and chronic bacterial infection has been shown to induce their formation in mice [112]. When the B cell-depleting anti-CD20 antibody rituximab was administered to two CF patients prior to receiving lung transplants, lymphoid aggregates were not disrupted and their role in CF lung pathogenesis could not be determined [113]. Although lymphoid follicle formation can be induced by bacterial lung infection [112] and their presence has been identified in end stage CF [54], further investigation into the role of lymphoid neogenesis in CF is needed to determine if they should be targeted therapeutically [114]. Possible therapeutics to disrupt lymphocyte aggregation in the airways could target the chemokines responsible for B cell recruitment. An abundance of C-X-C motif chemokine ligand 12 (CXCL12) and 13 (CXCL13) has been identified in lymphoid aggregates from the lungs of individuals with bronchiectasis and CF compared to the airway epithelium of control nonsmokers [112]. In the same study, IL-17A was found to be increased in the epithelium of bronchiectasis and CF subjects compared to controls [112]. These cytokines are important for the recruitment of B cells [115,116] and are potential targets for cytokine blockade therapy. Cytokine blockade is currently being evaluated for efficacy in other inflammatory diseases, for example through targeting of the pro-inflammatory mediators IL-17 and tumor necrosis factor α (TNFα) in rheumatoid arthritis [117].

## 3. Protein-Directed Therapies

No cure has yet been developed for CF, but an improved understanding of CFTR biogenesis and regulation has opened new opportunities for therapeutic development [118]. Extensive work has been done to identify and develop CFTR modulators that can correct and potentiate the function of the mutated protein. Two major groups of these drugs include CFTR potentiators, such as ivacaftor, and correctors, such as lumacaftor and tezacaftor. These drugs used as single or combination therapies have shown efficacy in restoring lung function and other core clinical features of CF, and have been reviewed extensively [119,120,121,122]. However, drug development is complicated by a recent observation that CFTR correctors differentially affect expression of the N- and C-halves of the protein [123]. New approaches acting independently of CFTR will likely be needed. A recent study by Gianotti et al. investigated the use of small molecule anionophores to facilitate exchange of chloride and bicarbonate as a way to circumvent CFTR deficiency [124]. Another recent study has suggested the use on amphotericin to generate small unselective ion channels can modulate CF-related host defense defects [125].

The need for alternative therapies is reinforced by observations that managing long term inflammation still remains a challenge despite implementation of CFTR modulator therapies [126]. Continued development of innovative protein-directed therapies could yield new breakthroughs in treating CF lung disease. For example, Reihill et al. demonstrated that inhibition of proteases that activate signaling through the epithelial sodium channel in primary CF airway epithelial cells reduces channel activation and fluid absorption, which would possibly stimulate improved airway hydration and mucociliary clearance if used therapeutically as a CFTR-independent treatment [127]. These channel-activating serine proteases are secreted in great quantity by neutrophils in the CF airways where they acquire a pathological phenotype [27,128]. Neutrophil-directed protein-targeting therapies are a major area of unmet need given their broad roles in CF [17,18]. A recent report shows the potential breakthroughs of neutrophil-directed therapies by demonstrating that inhibition of the nucleotide-binding oligomerization domain-like receptors (NOD)-like receptor protein 3 inflammasome in a murine model of CF promotes clearance of *P. aeruginosa* and resolution of airway inflammation via reduced IL-1β production [129]. Targeting mechanisms of neutrophil exocytosis, as well as the activity of proteases, could offer new anti-inflammatory treatment options. However, the effect of CFTR modulators on macrophage function is also an area of interest and intriguing reports have been published on opportunities for targeting macrophages in CF, as detailed below.

### 3.1. Neutrophil Exocytosis

The enormous impact of neutrophils on progression of lung disease in CF makes them an obvious target for new therapies. Given the role of neutrophil exocytosis in driving disease progression [130], due to the release of factors that promote bronchiectasis [33,131], inhibition of exocytosis specifically in neutrophils offers the potential to improve patient outcomes without adversely impacting normal immune functions. The antiprotease secretory leukoprotease inhibitor was identified as having efficacy in reducing CF neutrophil exocytosis by interrupting calcium flux via reduced inositol 1,4,5-triphosphate production [132]. However, the effect on other essential neutrophil functions such as phagocytosis was not assessed. A potential breakthrough was published by Johnson et al. several years ago with the identification of neutrophil exocytosis inhibitors (nexinhibs). The group described several small molecules that inhibit the interaction between Rab27a and JFC1, which are key regulators of exocytosis. These compounds were shown to potently inhibit exocytosis of the primary (azurophilic) granules without reducing the ability to phagocytose and kill microbes. Additional functions including neutrophil extracellular trap formation (NETosis) were also unaffected [33]. Interestingly, blood neutrophils from CF patients have demonstrated reduced propensity for exocytosis compared to those of healthy controls [133], although more recent studies have demonstrated high exocytic activity of neutrophils in an in vitro CF airway model [134] and those from the airways of CF patients [130]. In this study it was found that treatment with the CFTR modulator ivacaftor increased neutrophil exocytosis to levels comparable to that of healthy control cells through activation of Rab27a [133]. These findings are important to consider as implementation CFTR modulatory therapies has failed to reverse chronic airway inflammation despite some recovery of lung function [15].

### 3.2. Neutrophil Elastase

Exocytosis of the neutrophil granules releases their destructive cargo into the extracellular environment where they can damage host tissue. A major example of this is NE, which has attracted significant attention as therapeutic target over the past decade [37,135,136]. The interest in NE as a target has been reinforced by recent findings demonstrating that surface-bound NE on neutrophils is associated with disease severity [130,137]. In addition, common treatments for CF patients such as deoxyribonuclease (DNAse) [138] may increase NE-related lung pathology by promoting enzyme activity [139], further demonstrating the need for developing therapies directed toward inflammatory mediators. NE activity in CF sputum promotes an aggregation phenotype in *P. aeruginosa* demonstrated by antibiotic resistance and reduced invasiveness independent of biofilm-promoting mechanisms [140]. These findings demonstrate that rather than exerting bactericidal functions as NE normally does intracellularly, extracellular NE in CF can actually aid in establishing infection.

Several recent trials have demonstrated the efficacy of therapeutically targeting NE in CF lung disease. The small molecule NE inhibitor KRP-109, which was previously shown to potently suppress neutrophilic inflammation in a mouse model of pneumococcal pneumonia [141], demonstrated efficacy in reducing the degradation of mucins in sputum from CF patients and thus may be effective at improving mucociliary clearance [142]. Oral administration of the new NE inhibitor POL6014 was shown to effectively target the airways and inhibit NE activity in the sputum, with only low concentrations detected in plasma [143]. Further research is needed to identify the best compounds for selectively inhibiting NE with minimal off-target effects. Using sulfur fluoride exchange, Zheng et al. identified a novel inhibitor of NE, benzene-1,2-disulfonyl fluoride, and a derivative of this compound with even greater potency, 2-(fluorosulfonyl)phenyl fluorosulfate. Importantly, this inhibitor was not cross-reactive with the homologous protease cathepsin G [144]. This technique will facilitate discovery of new compounds that can selectively target proteins with pathological activity and minimize the risk for interrupting normal immune functions. Indirect targeting of NE may also prove efficacious. As discussed recently by Hunt et al., inhibition of microRNAs that target the antiprotease alpha-1 antitrypsin (A1AT) could also help counter the massive NE burden in the lung by promoting the function of one of its endogenous inhibitors [145]. microRNAs that target the mRNA transcripts of A1AT suppress production of this key antiprotease. Inhibition of these non-coding negative regulators could allow for increased production of A1AT to counteract the destructive effect of excessive NE release.

### 3.3. Matrix Metalloproteinases

MMPs are another class of proteases implicated in CF lung disease, produced not only by neutrophils but also by macrophages and epithelial cells [39]. MMP-9 is especially interesting in CF lung disease, since NE has been shown to cleave and activate pro-MMP-9 as well as degrade the tissue inhibitor of metalloprotease-1 [146]. MMPs have a potent ability to degrade extracellular matrix and MMP-9 is capable of inducing the release of matrikines such as proline-glycine-proline [147] and drive further neutrophilic inflammation in vivo (reviewed in [148]). MMP-9 has also been shown to inhibit wound healing of epithelial cells when exposed to bacterial infection [149]. The importance of this particular protease was reinforced in a report by Garratt et al. demonstrating that in early disease, measurement of MMP-9 in bronchoalveolar lavage fluid is an effective predictor of bronchiectasis, while MMP-1 and MMP-7 were minimally present and MMP-2 was possibly degraded by NE [131].

Although much attention is given to neutrophil-derived MMP-9, Averna et al. demonstrated that peripheral blood mononuclear cells from CF patients are highly predisposed to secrete MMP-9 compared to healthy donor cells. This was attributed to enhanced signaling through calpain and protein kinase Cα promoted by increased intracellular calcium due to loss of CFTR [150]. Although no trial has yet been conducted to specifically target MMPs in CF, Xu et al. treated exacerbating CF patients with doxycycline, an antimicrobial with anti-MMP activity, and demonstrated reduction of MMP-9 activity and notable clinical outcomes in a single-center clinical study [151]. In addition, Hentschel et al. conducted a study of CF patients who were administered intravenous antibiotics by assessing the presence of inflammatory mediators before and after treatment. They found that within a short time after treatment (median time of 6 days) there was a significant reduction in MMP-9 and several proinflammatory cytokines in nasal lavage fluid, but this decrease was not observed for NE [152].

These findings demonstrate that, as expected, not all inflammatory processes respond to treatments with the same kinetics and as such offer opportunities for personalized treatment approaches. In a rat model for liver ischemia-reperfusion injury, which exhibits protease-induced tissue damage, Wang et al. demonstrated that cleavage of vascular endothelial growth factor by MMP-9 exacerbated injury, and inhibition of MMP-9 rescued this defect [153]. However, vascular endothelial growth factor is implicated in progression of CF lung disease [154], so careful consideration must be given for therapies aiming to modulate protease function.

### 3.4. Effect of CFTR Modulators on Macrophages

Since the identification of mutated CFTR protein as the cause of CF in 1989, several classes of drugs have been developed to promote expression of functional protein [155]. These CFTR modulators were designed to primarily correct epithelial cell function in the airways, but recent studies have demonstrated the benefits of these therapies in immune cells, too. Macrophages are the dominant leukocyte subset in healthy airways and are essential for clearing pathogens and debris. As such, altered macrophage function in CF is detrimental to lung health [19,20]. A pro-inflammatory poise of macrophages in the CF lung was suggested by Tarique et al. to be directly linked to CFTR deficiency [7]. In that study, blood monocytes from CF patients were less able to differentiate to the anti-inflammatory M2 macrophage phenotype compared to healthy donor cells. Furthermore, inhibition of CFTR in healthy donor cells recapitulated the pro-inflammatory phenotype observed with CF monocyte-derived macrophages. A growing body of literature now suggests that CFTR offers opportunities for therapeutic targeting. Zhang et al. demonstrated that blood monocyte-derived macrophages from CF patients have decreased phagocytic capability and are more prone to apoptosis than cells from non-CF subjects. Interestingly, while cells obtained from CF patients on ivacaftor therapy exhibited restored phagocytosis of *P. aeruginosa* compared to untreated patients, lumacaftor/ivacaftor combination therapy worsened the phagocytic response in cells from both CF and non-CF subjects treated in vitro [8]. Conversely, a separate study showed that lumacaftor improved killing of *P. aeruginosa* while addition of ivacaftor abrogated this effect in a dose-dependent manner [4]. Further research in needed to understand the possible beneficial effects of single therapies on pathogen clearance but abrogation of this effect by combination treatments.

One possible explanation for the lack of bactericidal activity in CF macrophages is impaired calcium influx due to reduced signaling through transient receptor potential vanilloid 2. Expression of this channel was found to be reduced on CF macrophages and inhibition of CFTR function on non-CF macrophages impaired calcium influx [156]. Impaired production of reactive oxygen species has been postulated as another mechanism for the inability of CF macrophages to kill bacteria. Impaired activation of the NADPH oxidase complex necessary for reactive oxygen species production has been demonstrated in CF macrophages [157], and this defect in reactive oxygen species production has been linked to an inability to kill intracellular bacteria [158]. These findings demonstrate that in addition to restoring ion flux in airway epithelial cells, targeting expression of CFTR in macrophages can rescue macrophage function by acting through a variety of pathways. However, more recent studies have challenged these findings. Using surface-enhanced Raman spectroscopy-based nanosensors, Law et al. demonstrated that CF monocyte-derived macrophages do not have impaired acidification of phagolysosomes compared to those of healthy controls [11]. Furthermore, Leuer et al. investigated the phagocytic capacity of neutrophils and monocytes from venous blood of CF and healthy subjects and found no significant difference for either cell type [12]. Further research is needed to ascertain the effect of CFTR loss on immune cell function.

## 4. Nucleic Acid-Based Therapies

There are already many small molecule drugs currently in use to treat CF lung disease. However, a subset of patients with rare mutations does not have access to these drugs. Furthermore, patients that do have access to a complete regimen of medications still see a gradual decline in their health, punctuated by acute exacerbations, which necessitates the need for more effective treatment strategies. Nucleic acid therapeutics could be used to target CF disease pathogenesis at its root cause by exogenously supplying the wild-type transcript to enable expression of fully functional CFTR or modifying the genome or transcriptome. There are a number of excellent reviews that describe the various nucleic acid therapeutics currently being developed for CF [159,160,161]. Therefore, we will focus our attention on the oft underestimated immune hurdles that these therapeutics have to overcome in vivo to be effective. As most nucleic acid-based therapies will have to be administered into the lung likely through inhalation. To achieve optimal delivery one has to consider, aside from the physical barrier given by mucus accumulation in CF, the inflammatory component of the disease. Epithelial cells and immune cells, particularly those in the innate myeloid compartment, are wired to detect non-self proteins, lipids, and nucleic acids through pattern recognition receptors (PRRs) [162].

DNA-based therapeutics must not only reach the target cell, but they also have to pass the nuclear membrane, become transcribed, be processed as pre-mRNA, and be exported from the nucleus to the ribosomes. While DNA vectors to deliver full length wild-type CFTR have been tried in the past, even reaching phase II clinical trials with a small increase in FEV1 in the experimental group [163], one has to consider the downstream effects of therapeutic success in addition to the challenges of effective therapeutic delivery. In patients with CF, their immune system has been tolerized toward the mutated form of CFTR [164], therefore, effective delivery and translation of wild-type CFTR in these patients may result in the presentation of peptides which could result in T cell activation towards transfected epithelial cells.

Antisense oligonucleotides (ASOs) are single-stranded DNA or RNA oligonucleotides designed to bind RNA targets via Watson-Crick base pairing [165]. ASOs typically downregulate or degrade their targets by RNase H mediated cleavage of the complementary transcript or modulate alternative splicing of pre-mRNA by sterically blocking splicing enhancers [166,167]. Immune response profiling downstream of ASO administration has not been extensively conducted. Such profiling could potentially identify side-effects resulting from the activation of nucleic acid-sensing pathways and downstream inflammation. So far, ASOs at lower doses seem to be well-tolerated in humans overall. For example, two ASO therapeutics were recently approved for treating familial hypercholesterolemia, and nusinersen, a splice-switching ASO, was approved for use in patients with spinal muscular atrophy, and multiple ASOs were approved for Duchenne’s muscular dystrophy [168,169]. A small subset of mutations within the CFTR gene result in aberrant pre-mRNA splicing. Notably, for CF, eluforsen has recently completed phase 1b trials in 2017 and was deemed safe after 8 weeks of use in homozygous deltaF508 patients leading to an improvement in Cystic Fibrosis Questionnaire-Revised Respiratory Symptom Score was seen in the trial group, while it worsened in the placebo control group [170]. However, the drug was used for a relatively short period of time and only indirect markers of inflammation were measured. With high levels of inflammation occurring in the CF lung, the formation of RNA:DNA duplexes may activate intracellular pattern recognition receptor pathways such as cyclic-GMP-AMP synthase (cGAS) and stimulator of interferon genes (STING) or the ASO may have the potential to activate endosomal TLR pathways such as TLR-7, -8, or -9 [171,172,173]. Chemical modifications to ASOs that have the potential to antagonize immune activation without increasing off-target binding have been extensively reviewed elsewhere [174], however it is important that every ASO be optimized based on sequence specificity for its target, mechanism of action, deliverability to the target tissues and cells, and resistance to degradation.

Unlike ASO therapies which are targeted to very particular regions of given transcripts, application of exogenous mRNA encoding full-length wild-type CFTR has the potential to be “a one size fits all drug” applicable to patients with any CF disease-causing mutation. In CFTR^−/−^ mice, modified mRNA encoding CFTR was administered using chitosan–poly(lactic-co-glycolic acid) nanoparticles [175]. While increased levels of pro-inflammatory mediators TNFα or interferon alpha (IFNα) were not observed, other critical cytokines, such as keratinocyte chemoattractant (KC), an IL-8 homolog, or interferon beta (IFNβ), were unfortunately not measured. Interestingly, increased protein levels of CFTR were found in the treatment group. It should be noted however, that CFTR^−/−^ mice do not show the high levels of neutrophil infiltration and immune activation that the human functional counterparts do, making results obtained in this model somewhat difficult to extrapolate to patients. Nevertheless, other studies reported similar rescue of CFTR function when modified wild-type CFTR mRNA was administered to CFTR^−/−^ mice and primary cultured human airway nasal epithelial cells [176,177]. Similar to DNA-based therapies, however, RNA-based therapies can also trigger immune activation. The addition of modified nucleotides to the reaction, such as m6A, m5C and pseudouridine could drastically reduce activation of pattern recognition receptors TLR-3, -7, and -8 as measured in dendritic cells and primary plasmacytoid dendritic cells [178], and also prevent retinoic acid inducible gene I (RIG-I) activation although not impacting binding to transcripts [179,180]. Pseudouridine incorporation also reduced cleavage of a transcript upon 2′-5′-oligoadenylate synthetase (OAS) and ribonuclease (RNAse) L activation [181]. In sum, using modified nucleotides in the synthesis of mRNA allows for evasion of innate immune system PRRs and thus may reduce immunogenicity when used in vivo.

## 5. Conclusions

As exemplified by the development of CFTR-directed therapies, including approval of Trikafta^®^ in 2019 [119], the majority of recent clinical trials for CF have focused on incorporating these novel small molecule therapeutics into the drug regimen. Indeed, there are currently 69 interventional clinical trials (active or recruiting) in the US for CF (clinicaltrials.gov), a majority of those studies aims at modulating CFTR. A few trials are focused on inflammatory and infectious components of the disease (reviewed in [182]), three of which are in phase II. Of note, a clinical trial exploring the safety and tolerability of full length CFTR mRNA delivery to the lungs by inhalation is also ongoing (CFF.org/trials), possibly a sign of things to come with regards to upcoming CF gene therapy trials.

### Unmet Need for Immunotherapies

There is a plethora of candidate protein or nucleic acid-based therapies being developed that show promising results for either personalized medicine-based or one-size-fits-all approaches to curing CF disease. Although some have moved forward to the clinical testing phase most of these candidate therapies have not been tested in patients yet. A critical bottleneck in drug development is that existing CF animal models do not include all of the relevant barriers to entry nor fully recapitulate the peculiar immunological landscape in the lung of CF patients. Improved in vitro and animal models are therefore needed to further optimize not only candidate treatments, but also delivery methods to target specific airway cells. Further research into the complex interplay resulting in CF lung inflammation will provide important direction toward development of better treatments, either based on direct immunomodulation, or as adjunct therapies for nucleic acid delivery approaches. Moreover, particular attention should be directed to the mechanisms underlying the architecture of lung tissue, as chronic inflammation triggers tissue remodeling and repair mechanisms. To this end, the investigation of small molecules targeting both inflammation and tissue remodeling is of interest. However, while novel therapies have reduced toxicity, using drugs affecting a large variety of cell types may lead to increased side effects, particularly in CF, where patients are subjected to an already heavy therapeutic regimen. It will also be critical to assess use of these new therapies in combination with CFTR modulators, which have become the “new normal” in CF clinical management.

## Figures and Tables

**Figure 1 ijms-21-03331-f001:**
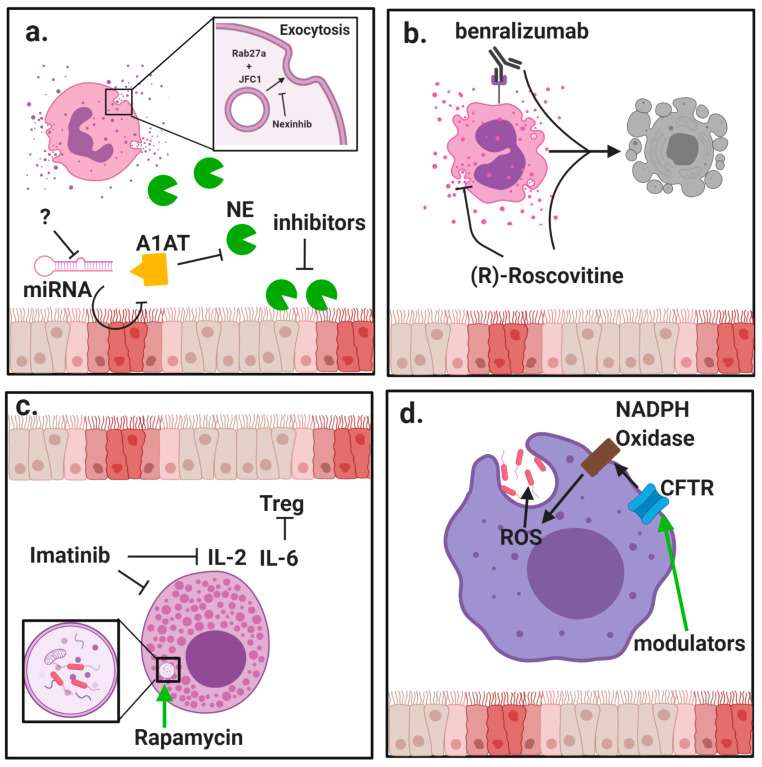
Modulation of innate immune cell function by protein-directed therapies. Innate immune cells, including granulocytes and macrophages, have important roles in the pathophysiology of cystic fibrosis (CF) lung disease. (**a**) Neutrophils are massively recruited to the airways in CF and fail to clear pathogens despite their highly inflammatory activity. Neutrophils rapidly exocytosis their granules, releasing destructive enzyme such as neutrophil elastase (NE) into the extracellular space. Alpha-1 antitrypsin (A1AT) is a crucial anti-protease in the lung but is overwhelmed by the burden of NE in advanced stages of disease. New potential mechanisms to counter NE-driven inflammation include inhibition of micro RNA (miRNA) against A1AT and novel NE inhibitors with increased potency. Specific inhibition of granule exocytosis without affecting other functions can be achieved by neutrophil exocytosis inhibitors (nexinhibs) via inhibition of the interaction between the two docking proteins Rab27a and JFC1. (**b**) Eosinophils are much rarer than neutrophils but they can have a potent role in comorbidities with CF such as allergic bronchopulmonary aspergillosis (APBA). (R)-Roscovitine may be effective in suppressing eosinophilic inflammation by blocking degranulation and inducing apoptosis. Benralizumab has also shown promise in promoting apoptosis in studies of asthma. (**c**) Mast cells are another rare granulocyte that present the opportunity for new therapies. Interleukin (IL)-6 blockade may relieve mast cell-mediated suppression of regulatory T cells (Tregs), and this may be achieved through use of imatinib which has been shown to suppress mast cell infiltration and secretion of inflammatory cytokines. Induction of autophagy by rapamycin has also improved bacteria killing by mast cells. (**d**) Besides neutrophils, macrophages are the other major phagocyte in the lungs. CF macrophages have reduced ability to kill bacteria but the direct role of cystic fibrosis transmembrane conductance regulator (CFTR) deficiency is still debated. Use of CFTR modulators has shown some ability to restore bacteria killing, which could be due to restored activity of nicotinamide adenine dinucleotide phosphate (NADPH) oxidase and increased generation of reactive oxygen species (ROS).

**Figure 2 ijms-21-03331-f002:**
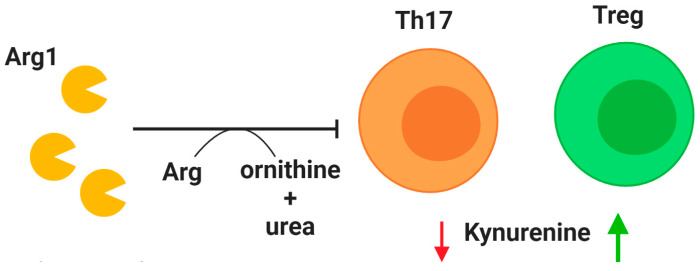
Modulation of T cell activity. **Arginase 1** (Arg1) is secreted by neutrophils upon granule exocytosis and has a potent inhibitory effect on T cells in the lung by cleaving the essential amino acid arginine (Arg) to produce ornithine and urea. A deficiency of regulatory T cells (Tregs) in CF patients may be explained by reduction of indoleamine 2,3-dioxygenase (IDO). Mouse models have demonstrated that correction of IDO deficiency by administration of kynurenines can rectify T cell imbalance by promoting Treg populations. Decreased availability of kynurenine (red arrow) reduces T helper type 17 (Th17) populations, while increased availability (green arrow) promotes Treg phenotype.

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
