# Peer review of "Immunomodulation in Cystic Fibrosis: Why and How?"

_ijms, 2020, doi:10.3390/ijms21093331_

Round 1

Reviewer 1 Report

Giacalone et al. submitted a review article “Immunomodulation in cystic fibrosis: why and how?” to IJMS.

I have only a minor comment as follows.

This review article very well summarizes current treatments used to target immune cells in the lungs, and highlight potential benefits and caveats of novel therapeutic strategies: specially the characterization of immune cell function in CF patients. However, I found little description on the direct relationship between CFTR mutation and immune cell function. It is much better for readers to understand the pathophysiology if the authors would describe the molecular mechanism on how disorders in immune cell function described in this review article are caused by CFTR mutation.

Author Response

We would like to thank the editors and reviewers for their critiques and acceptance of the manuscript,
and for the opportunity to submit a revised version. The comments provided were helpful and we feel
that they have improved the manuscript. We present a point-by-point response below and indicate
changes that were made in the manuscript. Within the main document, deletions are marked as red
struck-out text, and additions as red text.

REVIEWER 1

C1 - This review article very well summarizes current treatments used to target immune cells in the
lungs, and highlight potential benefits and caveats of novel therapeutic strategies: specially the
characterization of immune cell function in CF patients. However, I found little description on the direct
relationship between CFTR mutation and immune cell function. It is much better for readers to
understand the pathophysiology if the authors would describe the molecular mechanism on how
disorders in immune cell function described in this review article are caused by CFTR mutation.
R1 - We have addressed this topic in the introduction section on page 1, lines 31-42.

Reviewer 2 Report

The manuscript entitled "Immunomodulation in cystic fibrosis: why and how?" inherited about a systemic revision of white cell role in the CF syndrome is well written and suitable for pubblication after minor revisions.

  • could the authors bbetter describe in the text the potential mechanism based on miRNA adoption that could be target NE enzyme ?
  • The authors report that T cell downregulation represent a key factor in the pathogenesis of CF. In relation to this point, could they suggest if a ri activation of the conventional  T cell pathway may reduce CF sympthoms?
  • In the conclusion section, please, coudl the authors report if clinical trials are on going to estabilish a new theraeputic strategy for CF patient adminitration?
  • Today several mutations in CFTR genes are described. Could the authors define if one or more of these alterations may represent a prognostic factor in relation to a new therapeutic startegy?
  • In the conclusion, the authors suggest that a modulation of immune system may contribute to restore the cllinical parameters of CF patients. In relation to this point, could the authors express their opinion of a new potential approach based on administration of novel targeted molecules able to simultaneously inhibit infiammation and the angiogenesis (nintedanib like)? 

Author Response

We would like to thank the editors and reviewers for their critiques and acceptance of the manuscript,
and for the opportunity to submit a revised version. The comments provided were helpful and we feel
that they have improved the manuscript. We present a point-by-point response below and indicate
changes that were made in the manuscript. Within the main document, deletions are marked as red
struck-out text, and additions as red text.

REVIEWER 2

C2 - could the authors better describe in the text the potential mechanism based on miRNA adoption
that could be target NE enzyme ?
R2- we added a sentence expanding on targeting of NE enzyme on page 9, lines 41-44.

C3 -The authors report that T cell downregulation represent a key factor in the pathogenesis of CF. In
relation to this point, could they suggest if a reactivation of the conventional T cell pathway may
reduce CF symptoms?
R3- We expanded on this point on page 7, lines 7-11.

C4 - In the conclusion section, please, could the authors report if clinical trials are on going to
establish a new therapeutic strategy for CF patient administration?
R4- We added a short paragraph in the conclusion section to include current clinical trials in CF, page
12, lines 30-37.

C5 - Today several mutations in CFTR genes are described. Could the authors define if one or more
of these alterations may represent a prognostic factor in relation to a new therapeutic strategy?
R5- We addressed this point on page 1, lines 38-39.

C6 - In the conclusion, the authors suggest that a modulation of immune system may contribute to
restore the clinical parameters of CF patients. In relation to this point, could the authors express their
opinion of a new potential approach based on administration of novel targeted molecules able to
simultaneously inhibit inflammation and the angiogenesis (nintedanib like)?
R6- We addressed this comment in the conclusion section, pages 12-13.